# FABRIC: Personalizing Diffusion Models with Iterative Feedback

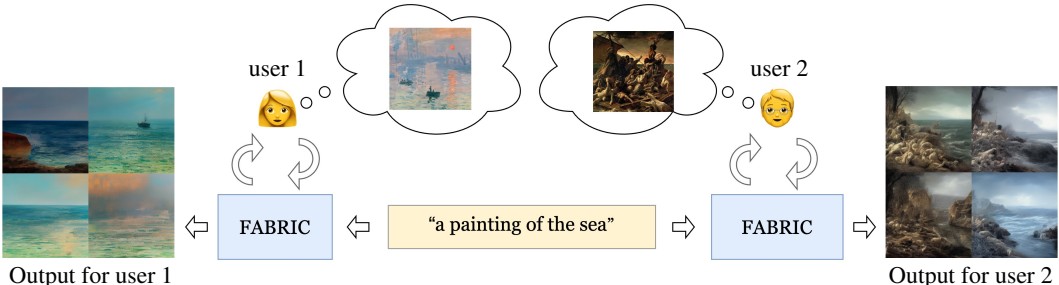

Figure 1: Illustration of the proposed approach. FABRIC generates images based not only on text prompt, but also on user preferences expressed during multiple rounds of generation.

## Abstract

In an era where visual content generation is increasingly driven by machine learning, the integration of human feedback into generative models presents significant opportunities for enhancing user experience and output quality. This study explores strategies for incorporating iterative feedback into the generative process of diffusion-based text-to-image models. We propose FABRIC, a training-free approach, which exploits the self-attention layer present in the most widely used diffusion models to condition the diffusion process on a set of feedback images. To ensure a rigorous assessment of our approach, we introduce a comprehensive evaluation methodology, offering a robust mechanism to quantify the performance of generative visual models that integrate human feedback. We show that generation results improve over multiple rounds of iterative feedback through exhaustive analysis, implicitly optimizing arbitrary user preferences. The potential applications of these findings extend to fields such as personalized content creation and customization.

## 1 Introduction

The field of artificial intelligence (AI) has witnessed a surge in interest in generative visual models, primarily due to their transformative potential across a myriad of applications, encompassing content creation, customization, data augmentation, and virtual reality. These models leverage advanced deep learning methodologies, such as GAN (Goodfellow et al., 2014) and VAE (Kingma & Welling, 2022), to generate high-fidelity and visually compelling images from given inputs or descriptions (Brock et al., 2019; Razavi et al., 2019). The significant advancements in generative visual models have catalyzed the exploration of novel possibilities in the realms of computer vision, natural language processing, and human-computer interaction (Radford et al., 2016).

Diffusion models, in particular, have emerged as a powerful tool in the field of image synthesis, often delivering results that are comparable to, or exceed those produced by GANs and VAEs (Ho et al., 2020; Rombach et al., 2022). These models are characterized by their ability to generate a diverse array of visually coherent images, while demonstrating superior stability and reduced mode collapse during the training phase (Song et al., 2021). Being conditioned on natural language or images, their variety can easily be controlled in arbitrarily many different ways, making them exceptionally useful for human image creation. This has led to their widespread adoption among researchers investigating the frontiers of generative visual modeling. Moreover, the utility of diffusion models extends beyond

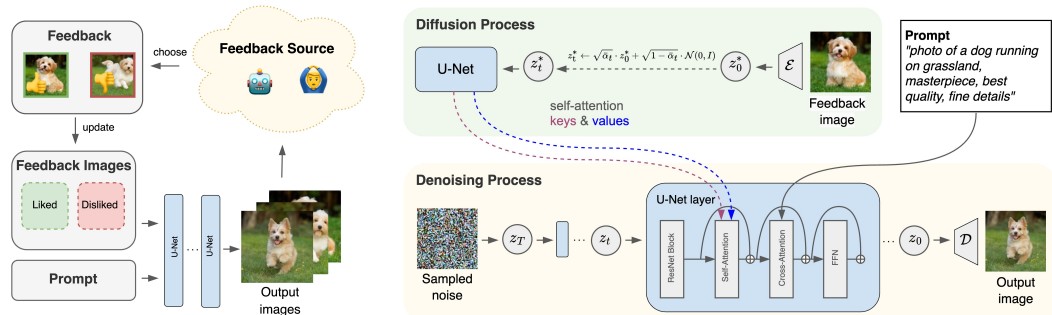

(a) Depiction of generating images with multi-round feedback.

(b) Details of FABRIC's method for conditioning the generation of latents $z_{t+1}$ based on one feedback image and one prompt.

Figure 2: Illustration of the proposed approach. FABRIC improves generated results by incorporating user feedback through an attention-based conditioning mechanism.

image synthesis, finding applications in various other domains such as inpainting, super-resolution, and style transfer (Chan et al., 2020; Park et al., 2019).

Text conditioning serves as a crucial component of generative visual models, enabling them to synthesize images based on human-readable descriptions (Reed et al., 2016). However, despite the robust conditioning, artists often have to iteratively optimize their prompts and parameters to improve results and achieve the intended outcome. This iterative workflow can be time-consuming and is susceptible to user error, as some prompting techniques work better than others and behave differently depending on the model. Still, users possess the ability to readily evaluate the quality of the generated images, opening up the possibility of integrating sparse human feedback into the generative process to enhance the results and better align them with user preferences.

In this work, we focus on this iterative workflow and propose a technique based on sparse feedback that aims to aid in steering the generative process towards desirable and away from undesirable outcomes. This is achieved by using positive and negative feedback images (e.g. gathered on previous generations) to manipulate future results through reference image-conditioning. Simply voting images as good or bad after each generated image batch allows for iterative refinement of the generated images based on arbitrary feedback sources (including human feedback). An illustrative overview of this system is given in Figure 2. Our contributions are three-fold:

- We introduce FABRIC (**F**eedback via **A**ttention-**B**ased **R**eference **I**mage **C**onditioning) [1], a novel approach that enables the integration of iterative feedback into the generative process without requiring explicit training. It can be combined alongside many other extensions to Stable Diffusion.

- We propose two experimental settings that facilitate the automatic evaluation of generative visual models over multiple rounds by introducing different proxies to emulate human feedback in an automated fashion.

- Using these settings, we evaluate FABRIC and demonstrate its superiority over baseline methods on a variety of metrics, including feedback proxy scores. We also empirically quantify the quality-diversity trade-off, which is common in generative model alignment.

## 2 FABRIC

In order to alleviate the necessity of extensive prompt-engineering, which is a tedious and time-consuming part of current text-to-image models, we will now introduce our proposed method, FABRIC, which provides a way to incorporate multiple rounds of positive and negative feedback into the generative process. Specifically, we split the problem into three distinct sub-tasks: How can we

---

[1]The code is publicly available: `https://anonymous.4open.science/r/fabric-FE32/README.md`

manipulate the generative process in a way that conditions it on a reference image? How can we use reference-image conditioning in order to incorporate binary feedback? And how can we extend this feedback mechanism to an iterative setup?

**Attention-based Reference Image Conditioning.** FABRIC takes inspiration from a technique implemented in an update to the widely used ControlNet that introduces the ability to generate synthetic images similar to some reference (Zhang, 2023). Intuitively speaking, the mechanism *makes the model draw information from reference images* during the generative process by carefully injecting said information at each step. This is done by exploiting the self-attention module in the U-Net. Since the self-attention module's weights have learned to "pay attention" to other pixels in the image, adding additional keys and values from a reference image offers a way to inject semantic information. In order to be compatible with the generated image at timestep $t$ of the denoising process, the reference image $x_0^*$ is rescaled, encoded to its latent representation $z_0^*$ and partially noised up to the current timestep using the standard forward diffusion process Ho et al. (2020):

$$z_t^* \leftarrow \sqrt{\bar{\alpha}_t} \cdot z_0^* + \sqrt{1 - \bar{\alpha}_t} \cdot \mathcal{N}(0, I) \tag{1}$$

The keys and values needed to compute attention on the reference image are acquired by feeding $z_t^*$ through the U-Net of Stable Diffusion (Rombach et al., 2022) and storing the keys and values (or equivalently: the hidden states) of the self-attention module of each layer, which is given by the following formula:

$$\text{Attention}(Q^*, K^*, V^*) = \text{softmax}\left(Q^* K^{*\top}\right) V^* \tag{2}$$

where $Q^*, K^*, V^*$ are linear projections of the hidden state of the reference image.

Then, for a particular U-Net denoising step of the user prompt and the current partially denoised latent image $z_{t+1}$, the stored reference keys and values are appended in the self-attention block of the respective U-Net layers, computing $\text{Attention}(Q, \hat{K}, \hat{V})$ with $\hat{K} = \text{concat}(K, K^*)$ and $\hat{V} = \text{concat}(V, V^*)$. This way, the denoising process attends to the reference image and ingests semantic information from it. By reweighting the attention scores, we additionally obtain control over the strength of the reference influence as follows:

$$\text{WeightedAttention}_w(Q, \hat{K}, \hat{V}) = \text{softmax}\left(Q\hat{K}^\top + \log(W)\right)\hat{V} \tag{3}$$

where $W = \text{concat}\left(\mathbf{1}_{N \times N}, w \cdot \mathbf{1}_{N \times C}\right)$ with $N$ denoting the number of queries and $C$ denoting the number of feedback keys and values. An equivalent, more intuitive formulation of weighted attention is given in Eq. 7, App. A.2. While the keys and values of the generated latents $z_{t+1}$ have weight 1, the attention features (keys and values) of the reference latents $z_{t+1}^*$ can have a weight $w \leq 1$ modulating the conditioning strength.

**Incorporating Feedback in the Generation Process.** Armed with a technique to inject reference images into the generative process, we now extend this to multi-round binary feedback. Given a set of liked and disliked images, we precompute $K^*$ and $V^*$ as outlined above in a separate U-Net pass for every feedback image using the empty prompt (null prompt) as text condition. Note that a user-defined prompt at this point could be used to guide the feature extraction process in a more fine-grained manner. Also, as the attention injection is independent of the reference shape, one could also use image cutouts as references allowing more fine-grained control of feedback.

The attention features of liked and disliked images are injected into the conditional and unconditional U-Net pass respectively (see Classifier-Free Guidance) (Ho & Salimans, 2022) guiding the image generations to be more similar to the positive and less similar to the negative feedback.

We modulate the feedback by using weighted attention with $w$ depending on the feedback label and on the current timestep in the denoising process, all of which is configurable via hyperparameters. Namely, we configure the overall feedback weight (feedback strength), at which denoising steps to use feedback (feedback schedule), and how strong negative feedback is relative to positive (negative feedback scale).

Note that precomputing the attention features requires an additional U-Net forward pass for each image, roughly increasing computation time linearly in the number of feedback images. Further, concatenating $K^*$ and $V^*$ in the self-attention layer also increases the compute and memory requirements linearly in the number of feedback images.

| Name | SD version | Checkpoint | LoRA |
|------|-----------|------------|------|
| Baseline | 1.5 | stable-diffusion-v1.5 | ✗ |
| Dreamlike Photoreal | 1.5 | dreamlike-photoreal-2.0 | ✗ |
| HPS LoRA | 1.5 | dreamlike-photoreal-2.0 | HPS LoRA |

Table 1: Stable Diffusion models used during evaluation

**Extending to Iterative Feedback.**   Given the ability to incorporate positive and negative feedback, the extension to multiple rounds is straightforward: In the first round, we generate images without feedback (or optionally with user-provided seed-feedback images). From those images, a set of *liked* and *disliked* images is selected and added to the feedback. How the feedback is obtained is arbitrary, although we envision either a human user or a separate vision model being the feedback source. A new batch of refined images is generated using the updated feedback and the process is repeated. The full procedure is given as pseudocode in Appendix A.2.

## 3 EVALUATION

### 3.1 MODELS AND BASELINES

Since the proposed approach is training-free and model-agnostic, it is applicable to any model with self-attention modules (including any Stable Diffusion model). We conduct experiments with FABRIC applied to three baselines, all of which are based on Stable Diffusion 1.5, including the base model[2], a fine-tuned model called *Dreamlike Photoreal*[3] and a Low-Rank Adaptation (LoRA) based on Human Preference Score (HPS) (Wu et al., 2023b). An overview of the all models taken into account can be found in Table 1.

Since, to the best of our knowledge, there doesn't exist a method designed to incorporate iterative feedback gathered over multiple rounds, we compare the proposed method to standard Stable Diffusion models in the following manner. First, we run Stable Diffusion (either of the models mentioned above) $N$ times, each with a different seed, generating $N$ batches of images. Then, we collect the desired evaluation metrics for the generated batch at each round and we use this values to perform quantitative comparisons with FABRIC.

It is important to note that while we may add images from each round as feedback to future rounds for our method, the baselines do not have a mechanism to incorporate feedback into future generations. Therefore, the baseline models generate images independently in each round without taking previous rounds into consideration.

### 3.2 METRICS

**Preference Model**   In order to automatically estimate user preference we use the PickScore introduced in Section 5 as a proxy score for general human preference. Note that we use PickScore over HPS, which is another preference model, due to better alignment with human preference.

**CLIP Similarity to Feedback Images**   To assess the effectiveness of incorporating feedback into the generation process, we compute the CLIP similarity between the generated and the previous positive and negative feedback images.

For a generated image $x$ we compute the average CLIP-similarity to feedback images $y_{\text{pos}}^{(1)}, \dots, y_{\text{pos}}^{(k)}$ or $y_{\text{neg}}^{(1)}, \dots, y_{\text{neg}}^{(l)}$. Specifically, let $\text{CLIP}(x, y)$ denote the cosine similarity between CLIP-embeddings of $x$ and $y$. Then the positive CLIP similarity is defined as follows:

$$s_{\text{pos}}(x) = \frac{1}{k} \sum_{i=1}^{k} \text{CLIP}(x, y_{\text{pos}}^{(i)}) \tag{4}$$

The negative similarity $s_{\text{neg}}(x)$ is defined analogously.

---

[2]`https://huggingface.co/runwayml/stable-diffusion-v1-5`
[3]`https://huggingface.co/dreamlike-art/dreamlike-photoreal-2.0`

**In-Batch Image Diversity** In the process of steering the generation of images toward user preferences it is crucial to balance exploration (offering diverse image options for user selection) against exploitation (generating images aligned with previous feedback). To quantify the trade-off between these two factors across different rounds, we introduce a metric called In-Batch Diversity.

For a batch of images $x_1, \ldots, x_n$, we define the in-batch diversity based on the average CLIP-similarity between images in the batch. Specifically, with CLIP similarity defined as above, the In-Batch Diversity IBD is defined as follows:

$$\text{IBD}_{\text{CLIP}}(x_1, \ldots, x_n) = 1 - \frac{2}{n(n-1)} \sum_{i=2}^{n} \sum_{j=1}^{i-1} \text{CLIP}(x_i, x_j) \tag{5}$$

where $\frac{n(n-1)}{2}$ is the number of elements in the upper triangular cosine similarity matrix.

## 4    EXPERIMENTS

In order to evaluate the capabilities of our model, we designed two experimental settings, each employing a different criterion for selecting feedback images during rounds. In Section 4.1 we present a Preference Model-Based approach, where we select liked and disliked images using a preference score. In Section 4.2 we present a Target Image-Based approach, where feedback selection relies on the similarity with a target image provided at the start of the experiment (but not used as feedback directly). While we deem these two experiments to be fairly representative of a realistic workflow, we note anecdotally that a combination of prompt iteration and feedback via FABRIC tends to be a consistent, intuitive way to get high-quality results with minimal effort, being more natural and more effective than using either method on its own.

Both the experiments use a batch size of $4$ and iterate for 3 rounds (resulting in 2 rounds with feedback). After each round, one image is selected as liked and another one as disliked. We empirically find that a feedback strength of $w = 0.3$ strikes a good balance between incorporating feedback while preserving diversity (Sec. 4.3). Unless specified otherwise, we use $w = 0.3$ for positive feedback and $50\%$ of the positive weight for negative feedback (i.e. $w_{\text{neg}} = 0.15$ if $w_{\text{pos}} = 0.3$). We also find qualitatively that using feedback up until the last step can sometimes give worse results, so we only incorporate feedback during the first $80\%$ of denoising steps. Unless denoted otherwise, FABRIC is applied on top of Dreamlike Photoreal. More details on generation parameters are given in Appendix A.3.

### 4.1    PREFERENCE MODEL-BASED FEEDBACK SELECTION

The first experiment aims to evaluate FABRIC by assessing how humans generally perceive generated images through a universal preference score. Therefore, this experiment operates under the assumption that all humans have the same preference when providing feedback, which in general may not always hold and is challenged in the second experiment.

In practice, the experiment is conducted as follows. First, a random set of $1000$ prompts is sampled from the HPS dataset (Wu et al., 2023b). Starting with no initial feedback, we simulate a 3-round interaction between the model and the user. In each round we use the PickScore as a proxy for human feedback[4], select the highest/lowest-scoring image and add it to the positive/negative feedback respectively. For each batch of generated images we measure the PickScore as well as the average CLIP similarity to both positive and negatives feedback.

The results from our experiments are depicted in Figure 3. In Figure 3a we address the concern that simply running multiple rounds will tend to produce increasingly better results by pure chance (more samples have a better chance of getting lucky), which may lead to an increase in the maximum PickScore. FABRIC outperforms each respective baseline in this regard, and FABRIC (without HPS LoRA) even surpasses HPS LoRA, a model specifically trained to maximize human preference. In Figure 3b we observe that starting from the second round our method outperforms its respective baseline not only in terms of maximum PickScore but also in terms of the average and minimum

---

[4]In earlier experiments we used HPS and found similar results, namely that FABRIC also surpassed HPS LoRA on that metric.

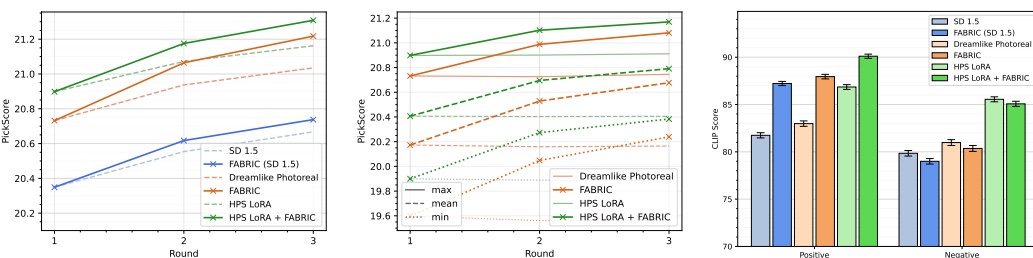

(a) Highest PickScore over all previous rounds ("best image so far")

(b) Minimum, mean and maximum PickScore in each round

(c) Similarity to pos./neg. feedback images in round 2

Figure 3: Results of preference model-based feedback selection

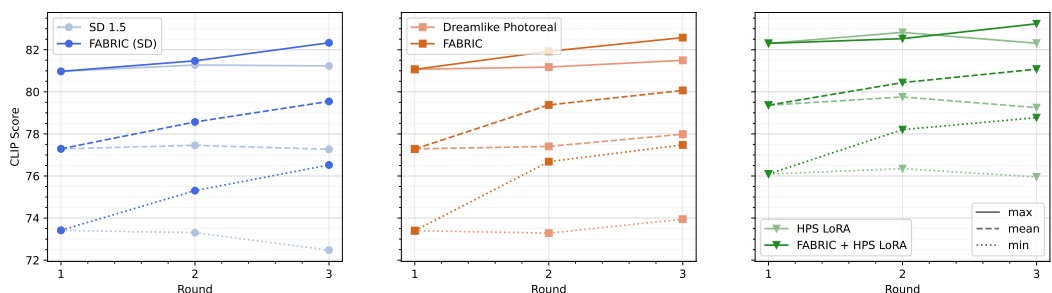

Figure 4: Results of target image-based feedback selection.

value, indicating an overall enhancement in the quality of generated images. In Figure 3c we evaluate the similarity of generated images to positive and negative feedback. We observe that even after just one round, the CLIP similarity score is higher for the positive samples and lower for the negatives compared to the respective baseline. This confirms that FABRIC effectively conditions the direction of generation based on the provided feedback.

## 4.2 TARGET IMAGE-BASED FEEDBACK SELECTION

In this experiment, we challenge the assumption from the previous setup that all humans have the same preference according to which they select liked/disliked images. Instead, we assume that the user has some target image in mind, some imagined picture that they would like to express. To reflect this scenario, we manually gather a dataset of prompt-image pairs from the AI-art-sharing platform *prompthero.com* where users post their favorite generations along with the prompt that was used to generate it. Due to this survivorship bias, we assume that shared images express the respective user's creative vision for the given prompt or, more generally, correspond to the user's preference. We manually clean the prompts, removing any UI-specific syntax (e.g. prompt-weighting).[5] The dataset consists of 135 prompt-image pairs and is provided in our public GitHub repository.[6]

The experiment setup is analogous to the preference-model based setup, with the exception of the feedback proxy: This time we select feedback images based on the highest/lowest CLIP-similarity to the target image associated with the prompt. We compare the baselines with FABRIC in terms of CLIP similarity to the target image in Figure 4. We find that FABRIC outperforms each respective baseline, improving best-case and significantly improving average and worst-case outcomes in rounds 2 and 3. Especially the per-round minimum similarity drastically improves over the baseline as soon as feedback is introduced. Appendix A.5 contains a more detailed analysis about the effect of the feedback schedule on this task.

We encourage the reader to also look at the qualitative results from this experiment in Appendix A.1, as they are very illustrative of how FABRIC works.

---

[5]https://github.com/AUTOMATIC1111/stable-diffusion-webui/wiki/Features#attentionemphasis

[6]https://anonymous.4open.science/r/fabric-FE32/README.md

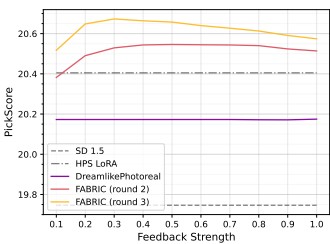
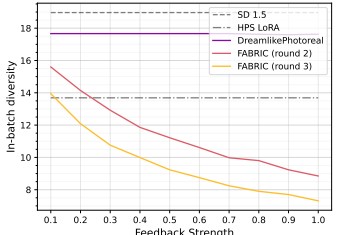
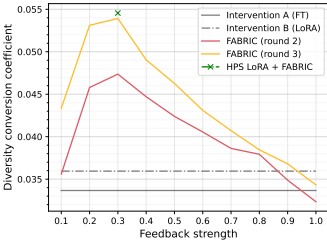

(a) Average PickScore of generated images (higher is better)

(b) In-batch image diversity (higher is better)

(c) Diversity trade-off efficiency (higher is better)

Figure 5: The effect of feedback strength on generated images. Increasing feedback weight lowers diversity, which adversely affects exploration. Hence, PickScore is optimal for intermediate strengths.

## 4.3 TRADING OFF DIVERSITY AND QUALITY

The trade-off between quality and diversity is omnipresent in the context of image generation. For example, fine-tuning a model on a small dataset of curated images can drastically increase the quality but usually comes at the cost of diversity in generated images. In the case of FABRIC, increasing the feedback strength produces images that more strongly adhere to the feedback while also decreasing diversity. In an ideal world, one would like to maximize both quality and diversity simultaneously, but maximizing one usually comes at the detriment of the other. To study the effect of feedback strength on this trade-off, we conduct the preference-model based experiment with varying feedback weights ranging from $0.1$ to $1.0$. The results in terms of PickScore and image diversity are illustrated in Figure 5. Note that the first round of FABRIC is equivalent to Dreamlike Photoreal, since no feedback has been added yet. We see the aforementioned trade-off in action, with the SD 1.5 base model having the highest diversity but lowest PickScore. Conversely, Dreamlike Photoreal and HPS LoRA trade some of the diversity for increased PickScore, with HPS LoRA having the highest score but lowest diversity out of the baselines. FABRIC exhibits similar behavior: While increasing the feedback strength strictly decreases in-batch diversity, the same does not hold for quality, which peaks around a strength between $0.2$ and $0.3$.

We also investigate the cost of increasing quality by asking the following question: How much quality can be gained by paying the same decrease in diversity for different models and feedback strengths? To quantify this trade-off, we estimate a conversion rate between diversity and PickScore by computing the linear least squares fit to the relation

$$\Delta\text{PickScore} = \alpha \cdot (-\Delta\text{IBD}_{\text{CLIP}}) \tag{6}$$

where $\Delta\text{PickScore}$ and $\Delta\text{IBD}_{\text{CLIP}}$ denote the change in PickScore and in-batch diversity respectively. The diversity conversion coefficient $\alpha$ denotes the rate at which sacrifices in diversity get converted to increased quality. We compare FABRIC with varying feedback strengths to two different interventions. *Intervention A*: Fine-tuning a model (comparing Dreamlike Photoreal to SD 1.5), *Intervention B*: applying the HPS LoRA to Dreamlike Photoreal. We find that fine-tuning has the lowest conversion efficiency, followed by the HPS LoRA. This could be either due to the fact that HPS LoRA is explicitly trained on maximizing human preference or due to parameter-efficient training preserving generalization abilities of the base model. FABRIC surpasses all baselines for all feedback strengths except $1.0$ where diversity and quality both suffer compared to lower strengths. Incidentally, applying FABRIC on top of HPS LoRA still has the same conversion efficiency as when applied to Dreamlike Photoreal, despite the already-diminished diversity, suggesting that the trade-off efficiency of FABRIC is robust with respect to different models.

## 5 RELATED WORK

**Textual Inversion and Style Transfer.** A popular method for personalizing text-to-image diffusion models is textual inversion (Gal et al., 2023; Ruiz et al., 2023), a technique for learning semantic text embeddings of a concept depicted in a set of images. This enables the synthesis of new images that express the learned concept. Textual inversion methods often require multiple images of the

same concept and additional training to learn the concept embedding. Null-text inversion (Mokady et al., 2022) is a very accurate inversion technique, enabling text-based real-image editing capabilities through pivotal inversion and null-text optimization. Xu et al. (2023) introduce a novel approach to condition on a user-provided reference image instead of natural language. A Semantic Context Encoder (SeeCoder) transforms the input image into semantic visual embeddings, which replace the text conditional inputs to a Text-to-Image (T2I) model. StyleDrop (Sohn et al., 2023) is able to synthesize images that adhere to a specific style. A T2I model is adapted to generate images in a user-provided style through iterative training with either human or automated feedback.

**Human Preference Modelling.** Recently, modeling human preferences in generative models experienced increased attention. In the domain of T2I models, initial work by Wu et al. (2023b) released a dataset of prompts and images, along with human preference annotations. The authors train a CLIP-based classifier (Radford et al., 2021) on human preference prediction, giving rise to the *Human Preference Score* (HPS), as well as a LoRA (Hu et al., 2021) of Stable Diffusion that significantly improves image quality in terms of HPS. They update their dataset in Wu et al. (2023a). Kirstain et al. (2023) introduce Pick-a-Pic, a large-scale dataset of user preferences for text-to-image generation collected from real users, containing 583,747 rankings and achieving state-of-the-art results on human preference prediction. Fan et al. (2023) proposes a policy gradient optimization method called DPOK for fine-tuning text-to-image diffusion models using online reinforcement learning, outperforming supervised fine-tuning techniques. Concurrent work by Deckers et al. (2023) integrate users preferences by navigating the prompt embedding space towards nearby variants, guided by gradients on a target metric, by user feedback, or by similarity to a reference image.

**Image Editing.** Instead of incorporating human preferences into newly generated images, capable image editing techniques can help an artist to modify existing images towards their preference. In Prompt-to-Prompt (Hertz et al., 2022), the cross-attention between pixels and the language prompt is changed to modify the image generation. Compared to this cross-attention injection, our method instead focuses on image-to-image self-attention of a diffusion model (see Section 2). Imagic (Kawar et al., 2023) instead interpolates between a text encoding optimized to recreate the reference image and the real target prompt to edit an image towards the target prompt. DiffEdit (Couairon et al., 2022) encodes an image to a timestep and decodes within a generated mask with a new prompt. One can also finetune a diffusion model (on Prompt-to-Prompt data) to follow editing instructions, as done by (Brooks et al., 2023). The authors condition on the reference image by concatenating it to the first convolutional layer in the diffusion model (as opposed to our training-free self-attention injection).

**Iterative Feedback.** The T2I workflow commonly consists of thinking of a prompt and iteratively improving it until the desired result is achieved. In order to alleviate some of this burden from the user, Tang et al. (2023) propose a zeroth-order optimization algorithm with theoretical guarantees for problems with ranking oracles and demonstrate its effectiveness by optimizing the initial seed for a fixed text prompt, using iterative human feedback as a ranking oracle.

To summarize, the literature of the field has predominantly focused on concept learning, style transfer from reference images, and modeling human preferences to fine-tune models accordingly. However, these methods require further training of the model and the iterative nature of how T2I models are used in practice has been left relatively unexplored.

## 6   LIMITATIONS AND FUTURE WORK

While FABRIC works well in our experiments, some limitations apply. We noticed that FABRIC is effective at constraining the generative distribution to a subset of preferable outcomes, but it struggles to expand the distribution beyond the initial text-conditioned distribution. Further, since we sample feedback images from previously generated images, the diversity tends to converge to a single mode for increasing numbers of feedback images and feedback strength. Albeit left unaddressed in our experiments, there are some simple remedies to combat this limitation: Varying the textual condition by adding character noise to the prompt (Deckers et al., 2023) or by automatically rephrasing the prompt via paraphrasing can help increase the diversity of the conditional distribution and by sampling feedback images not only from the model but also from an external feedback corpus Chen et al. (2022) can help to both delay convergence and increase the diversity beyond the initial distribution.

Another limitation lays in the way the feedback is collected. Currently, users can only provide binary preferences over images, which makes it impossible to only include certain aspects about a given image or to give scalar ratings. While left unexplored in this work, FABRIC is amenable to simple extensions that enable more detailed forms of feedback: By scaling the feedback strength according to the user's rating, continuous-scale feedback can be implemented. Further, while we fix the prompt for feature extraction from feedback images to the null prompt, a textual description of the feedback could be used instead in order to guide the feature extraction process. Finally, similarly to inpainting, attention masking could be used to constrain the feedback to certain parts of the image.

A distinct advantage of FABRIC is that it is easily extensible and orthogonal to most other Stable Diffusion variations, like fine-tuning, LoRA training or modifying the text conditioning (Xu et al., 2023), while achieving significant improvements on top of them (see FABRIC + LoRA). Also, extensions such as liking or disliking arbitrary masks (which is inherently supported by the attention injection) could increase the control of the artist. In addition, FABRIC provides a well-defined action space with various parameters that can affect the generated results. This opens up the possibility for performing Bayesian optimization on an arbitrary objective (e.g., direct user feedback or a score given by a preference model).

# 7    CONCLUSION

We present FABRIC, a training-free method that incorporates iterative feedback into the generation process of text-to-image models, leveraging attention-based reference image conditioning. Our experimental findings suggest that FABRIC is capable of implicitly optimizing a variety of objective functions such as human preference and similarity to a designated target image.

These objectives noticeably improve with more feedback rounds, demonstrating that FABRIC's effectiveness significantly exceeds merely sampling more images without providing additional feedback. Remarkably, even without training or hyperparameter tuning, FABRIC can outperform the HPS LoRA (a model explicitly trained to optimize human preference) on the relevant metric.

While FABRIC tends to trade diversity for increased quality, it does so in a controllable and predictable manner and provides a better trade-off efficiency than other quality-enhancing interventions such as fine-tuning or LoRA training. FABRIC still struggles with going beyond the model conditional distribution, requiring other methods to facilitate exploration.

The iterative setting is paramount to how generative visual models are used in practice. Despite this, recent research on aligning text-to-image models has left it largely unexplored. We believe that this study contributes towards the formation of a framework that aids in devising and evaluating methodologies intended to address this setting.

## ETHICAL IMPLICATIONS

Text-to-image models have the potential to make creative visual expression more accessible to everyone by allowing individuals without artistic skills or technical expertise to generate visually appealing content. Our proposed method aims to further enhance the accessibility and personalization of these models by incorporating user preferences into the image generation process. This is achieved through the utilization of positive and negative feedback, allowing users to provide natural and intuitive guidance based on prior images or previously generated ones.

By adopting our approach, users gain increased control over the generated content, promoting ethical usage. However, this heightened control also raises concerns regarding the potential for misuse or abuse of the system. To address these concerns, it becomes crucial for the community as well as society at large to establish clear guidelines regarding the legal and ethical utilization of such systems. By placing responsibility on individual users to ensure responsible and ethical usage, we can mitigate the risks and foster a positive and constructive environment for creative expression.

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

# A APPENDIX

## A.1 TARGET IMAGE-BASED FEEDBACK SELECTION EXAMPLES

We provide some example feedback-trajectories from the *Promthero* dataset in Figure 6. Liked and disliked images are selected by the highest and lowest CLIP similarity to the target image respectively (note that this does not necessarily agree with human judgement of similarity).

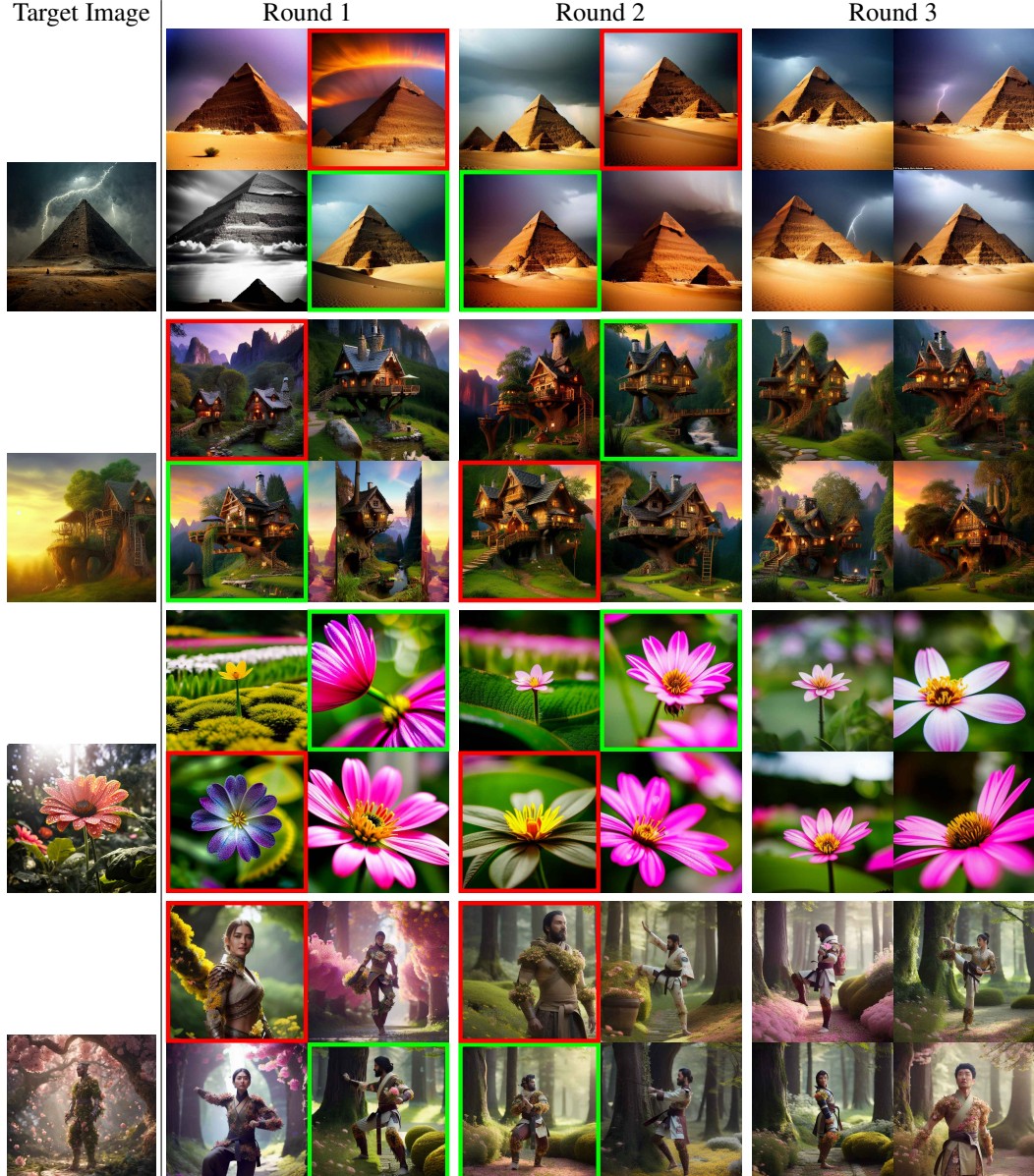

Figure 6: Examples of feedback rounds from our target-image-based experiment. Liked images are circled in green and disliked images in red.

## A.2 METHOD DETAILS

Here we provide the pseudocode of the FABRIC algorithm (see Algorithm 1). Weighted attention is defined for $Q \in \mathbb{R}^{N \times d}$ and $K, V \in \mathbb{R}^{N+C \times d}$ where $N$ and $C$ are the number of tokens in the generated image and reference images respectively:

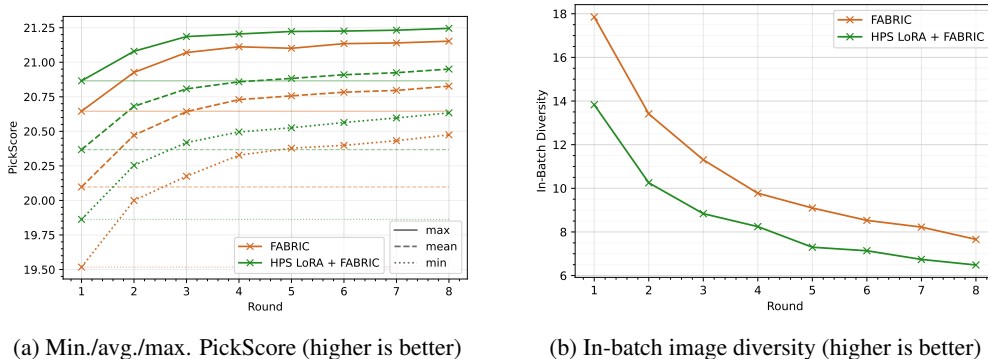

(a) Min./avg./max. PickScore (higher is better)  (b) In-batch image diversity (higher is better)

Figure 7: PickScores and in-batch diversity for up to 8 iterative feedback rounds. Average and worst-case outcomes keep improving throughout, but diversity also keeps decreasing.

$$\text{WeightedAttention}_w(Q, K, V) = \left( \frac{W \odot \text{softmax}\left(QK^\top\right)}{\left\| W \odot \text{softmax}\left(QK^\top\right) \right\|_1} \right) V \tag{7}$$

where

$$W = \begin{bmatrix} 1, ..., 1, w, ..., w \\ \vdots \\ 1, ..., 1, w, ..., w \end{bmatrix} \in \mathbb{R}^{N \times N + C}$$

i.e. the first $N$ columns of $W$ weight attention scores of query-key pairs from the generated image with $1$ and the last $C$ columns weight attention scores from reference keys with $w$.

## A.3 GENERATION PARAMETERS

Unless specified otherwise, the parameters specified in Table 2 were used for all baseline and FABRIC runs.

| | |
|---|---|
| Base model | DreamlikePhotoreal 2.0 |
| Batch size | 4 |
| Num. rounds | 3 |
| CFG scale | 6.0 |
| Denoising steps | 20 |
| Sampling schedule | Ancestral Euler |
| Feedback strength | 0.3 |
| Negative feedback scale | 50% |
| Feedback start | 0.0 |
| Feedback end | 0.8 |

Table 2: Default generation parameters that were used throughout the experiments

## A.4 CONVERGENCE BEHAVIOR FOR MANY ITERATIONS

To examine the behavior of FABRIC for increasing numbers of feedback images, we conduct additional runs for FABRIC and HPS LoRA + FABRIC using the preference model-based setup for 8 rounds. Due to computational limitations, only two runs were conducted with a sample size of 375 prompts. The results in Figure 7 shows that while the best-case image in each round converges after

---

**Algorithm 1** FABRIC: Feedback via Attention-Based Reference Image Conditioning

---

**Require:** Let $N$ be the number of feedback rounds, $n$ be the batch size of generated images in each round and *model* be a diffusion model capable of reference-conditioning.

1: **procedure** FABRIC
2:     pos, neg $\leftarrow$ [], []
3:     **for** $i \in \{1, \ldots, N\}$ **do**
4:         prompt $\leftarrow$ get_prompt($i$)
5:         images $\leftarrow$ [GENERATE(prompt, pos, neg) **for** $n$ **times**]
6:         $x_{\text{pos}}, x_{\text{neg}} \leftarrow$ get_feedback(images)                 ▷ we focus on one like and one dislike
7:         pos.put($x_{\text{pos}}$)
8:         neg.put($x_{\text{neg}}$)
9:     **end for**
10: **end procedure**
11: **function** GENERATE(prompt, positives, negatives)
12:     $z_T$ = initial_noise()
13:     **for** $t \in \{T, \ldots, 1\}$ **do**
14:         hiddens $\leftarrow \{\}$
15:         **for** $x_{\text{ref}} \in \{\ldots \text{positives}, \ldots \text{negatives}\}$ **do**
16:             $z_{\text{ref}} \leftarrow \sqrt{\bar{\alpha}_t} \cdot x_{\text{ref}} + \sqrt{1 - \bar{\alpha}_t} \cdot \epsilon_{\text{ref}}^{(t)}$         ▷ forward diffusion noising, $\epsilon_{\text{ref}}^{(t)} \sim \mathcal{N}(0, \mathbf{I})$
17:             $h \leftarrow$ PRECOMPUTEHIDDENSTATES($z_{\text{ref}}, t$)
18:             hiddens.put($h$)
19:         **end for**
20:         compute $w_{\text{pos}}^{(t)}$ and $w_{\text{neg}}^{(t)}$ according to the feedback settings
21:         $\epsilon_{\text{cond},t-1} \leftarrow$ MODIFIEDUNET($z_t, t$, get_positive(hiddens), $w_{\text{pos}}^{(t)}$)
22:         $\epsilon_{\text{uncond},t-1} \leftarrow$ MODIFIEDUNET($z_t, t$, get_negative(hiddens), $w_{\text{neg}}^{(t)}$)
23:         $z_{t-1} \leftarrow$ step_with_cfg($z_t, \epsilon_{\text{cond},t}, \epsilon_{\text{uncond},t}$)         ▷ ancestral Euler sampling in our case
24:     **end for**
25:     **return** $z_0$
26: **end function**
27: **function** PRECOMPUTEHIDDENSTATES($z, t$)
28:     hiddens $\leftarrow$ []
29:     **for** $i$-th layer in the Unet **do**
30:         apply ResNet block(s) to $z$
31:         hiddens.put($i, z$)                         ▷ just before self-attention
32:         apply self-attention to $z$
33:         apply cross-attention and FFN to $z$
34:     **end for**
35:     **return** $z$
36: **end function**
37: **function** MODIFIEDUNET($z, t$, hiddens, $w$)
38:     **for** $i$-th layer in the Unet **do**             ▷ for hiddens $= \emptyset$ this function is a standard U-Net
39:         $h \leftarrow$ hiddens.at($i$)
40:         apply ResNet block(s) to $z$
41:         $Q \leftarrow W_Q^{(i)} \cdot z$
42:         $K \leftarrow W_K^{(i)} \cdot \text{concat}(z, h)$
43:         $V \leftarrow W_V^{(i)} \cdot \text{concat}(z, h)$
44:         $z \leftarrow \text{WeightedAttention}_w(Q, K, V)$
45:         apply cross-attention and FFN to $z$
46:     **end for**
47:     **return** $z$
48: **end function**

---

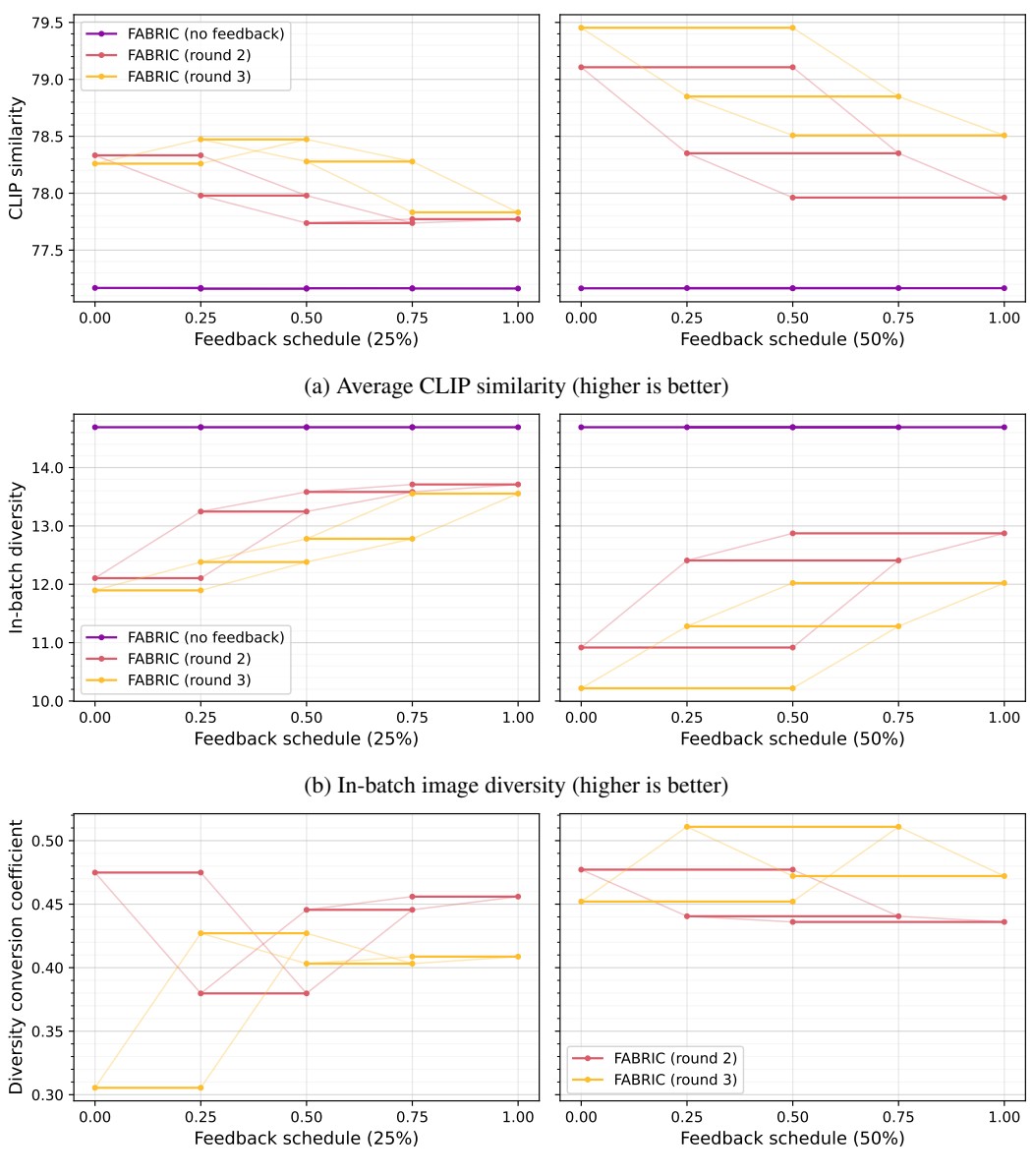

(a) Average CLIP similarity (higher is better)

(b) In-batch image diversity (higher is better)

(c) Diversity conversion efficiency (higher is better)

Figure 8: The effect of feedback schedule on generated images on diversity and CLIP similarity to target image. Applying feedback the first during the first half gives highest CLIP similarity to target, but feedback during thee middle half has the best trade-off efficiency.

4-5 iterations, the average quality and worst case outcomes keep improving up to round 8. However, diversity also continually decreases as more and more feedback images are added.

## A.5 FEEDBACK SCHEDULE

In addition to the experiments reported in the main part, we investigated the adaptation of the feedback schedule. The default configuration of FABRIC adds feedback in the first 80% of denoising steps. Here we investigate different schedules on the target image-based setup, i.e. using feedback during only 25% or 50% of denoising steps. We find that the feedback timing can make a big difference on both diversity and quality (in terms of similarity to the target image) and that our default setting

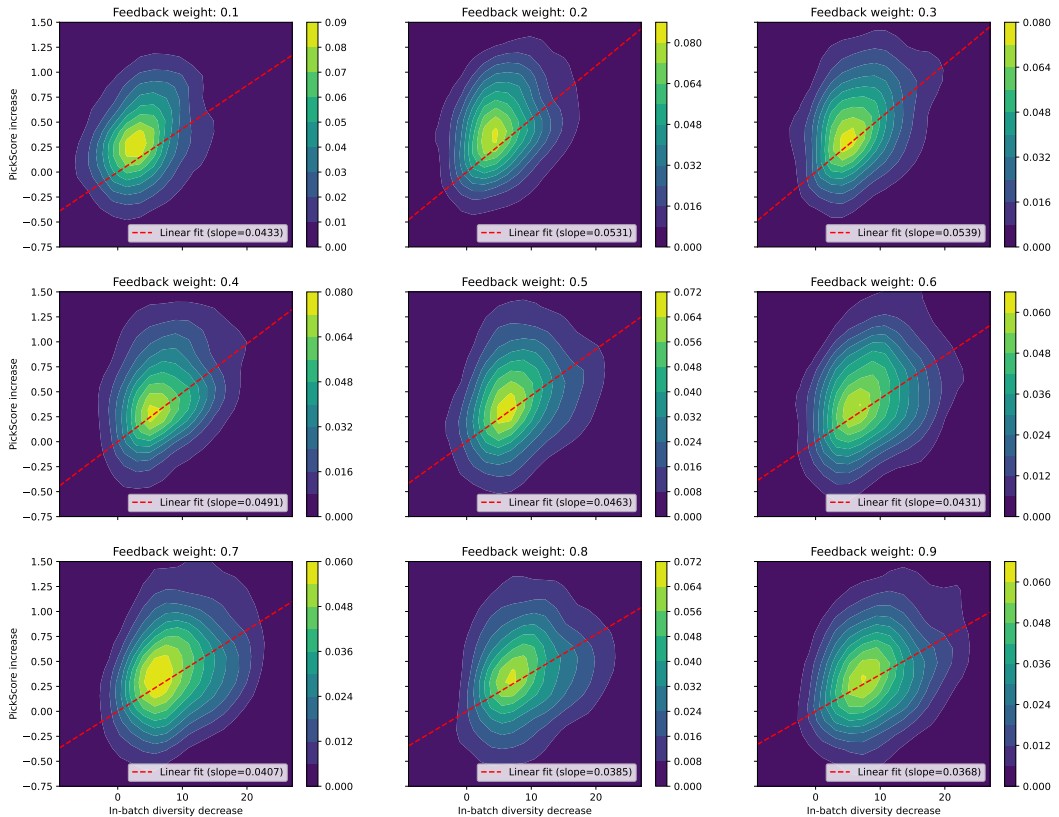

Figure 9: Density plots of the relation between increase in PickScore against decrease in diversity. While the distribution generally shifts to the right for increasing feedback strengths, the conversion slope has a local maximum around 0.3.

might not be ideal for every scenario. The results in Figure 8 show that using feedback in the middle half provides a good trade-off between diversity and CLIP similarity.

## A.6 QUALITY-DIVERSITY TRADE-OFF

Figure 9 shows the relationship between PickScore and in-batch diversity for different feedback strengths as well as the linear least squares fit for the conversion rate between quality and diversity. A single point is obtained by the mean PickScore and the in-batch diversity for any given round.

