# OpenReview forum: "FABRIC: Personalizing Diffusion Models with Iterative Feedback"
_ICLR.cc/2024/Conference — Submitted to ICLR 2024_

### Official Review · Reviewer_bHFH · 2023-10-24

**Soundness:** 3 good
**Presentation:** 3 good
**Contribution:** 3 good
**Rating:** 6
**Confidence:** 4

**Summary:**

In the growing field of machine learning-driven visual content generation, integrating human feedback can greatly enhance user experience and image quality. This study introduces FABRIC, a method that uses the self-attention layer in popular diffusion-based text-to-image models to condition the generative process on feedback images without additional training. Through a thorough evaluation methodology, the study demonstrates that iterative human feedback significantly improves generation results, paving the way for personalized content creation and customization.

**Strengths:**

1. Iterative Workflow: The research emphasizes an iterative process, allowing for continuous refinement and improvement of generated images based on previous feedback.
2. Dual Feedback System: By utilizing both positive and negative feedback images from previous generations, the method provides a balanced approach to influence future image results.
3. Reference Image-Conditioning: This approach manipulates future results by conditioning on feedback images, offering a dynamic way to steer the generative process.
4. Enhanced User Experience: By integrating human feedback into the generative models, the research ensures a more tailored and enhanced user experience in visual content generation.
5. Potential in Personalized Content Creation: The findings have significant implications for creating personalized visual content based on individual user preferences and feedback.

Overall, the paper introduces a robust and flexible method for refining machine-generated visual content through iterative human feedback, ensuring better alignment with user preferences.

**Weaknesses:**

1. Limited Expansion of Distribution: The method struggles to widen the distribution beyond the initial text-conditioned one provided by the model.
2. Feedback Loop Limitation: Since the feedback originates from the model's output, it creates a cyclical limitation where the model might only reinforce its existing biases.
3. Diversity Collapse: As the strength of the feedback and the number of feedback images increase, the diversity of the generated images tends to diminish. The images tend to converge towards a single mode that closely resembles the feedback images.
4. Binary Feedback System: The current feedback collection method only allows users to provide binary preferences (like/dislike) for the images. This limitation prevents users from providing nuanced feedback about specific aspects of an image.
5. Lack of Detailed Feedback: Users cannot specify which particular aspects of an image they appreciate or dislike. This restricts the model's ability to fine-tune its output based on detailed user preferences.

**Questions:**

See above

---

> ### Author Response · Authors · 2023-11-17
>
> We thank the reviewer for the accurate summary of our method and for the insightful and constructive comments. In the following, we would like to offer our perspective on some of the points raised.
>
> - **Limited Expansion of Distribution**
>
>   It is true that FABRIC does not necessarily expand the conditional distribution of generated images, especially when the feedback images are sampled from the same model, but this is generally the case for interventions that improve quality. Indeed, in order to improve quality it is necessary to constrain the distribution by emphasizing desirable and cutting out undesirable parts. Still, it is possible to reintroduce some diversity. For example, by adding feedback images from an external corpus in addition to the ones from the current generation, one can add more diverse images, guiding the generative process in new directions. This can even be automated by retrieving images from the external corpus that are similar to the prompt or existing feedback images (e.g. using CLIP similarity). Additionally, by adding random alphanumeric characters to the prompt it is possible to artificially increase the diversity of the generations [1]. We have added clarifying comments about this and the following two points to Section 6 of the manuscript.
> - **Feedback Loop Limitation**
>
>   Similar to the previous point, this is more a limitation of our experimental setup rather than a fundamental limitation of FABRIC. Indeed, it is possible to move the balance of the exploration-exploitation trade-off in the direction of the former with very simple extensions to FABRIC such as the addition of a retrieval corpus. More sophisticated methods might also take the prompt into consideration, by rephrasing the prompt for increased diversity or by iterating on the prompt and taking prior feedback into account, but we leave exploration of this issue to future work.
> - **Diversity Collapse**
>
>   We agree with the reviewer that in our experimental setup FABRIC clearly suffers from diversity collapse. However, we would like to point out that this is not necessarily the case in a general setup. In order to automate the process of evaluation, we were only giving images from the ones generated during previous rounds as feedback images. There, naturally (and hopefully), the process has to converge to a result very similar to the target image. In practice, the user can select feedback based on a variety of criteria, hence producing more diverse feedback images and (anecdotally) more diverse results. Additionally, we would like to emphasize that of all the methods we investigated, FABRIC was able to elicit the highest increase in quality metrics for the lowest decrease in diversity (see Section 4.3 on the quality-diversity trade-off), comparing favorably to both finetuning and LoRA training.
> - **Binary Feedback System**
>
>   The binary nature of feedback is to a large degree imposed by the binary nature of CFG. Still, scalar ratings can be approximated with minor modifications to FABRIC: One could choose different feedback strengths based on the rating, with the strongest rating receiving the highest feedback strength and the weakest rating having a feedback strength of 0. While we did not evaluate this extension, it is easy to implement on the application side. The main reason why this was excluded from the evaluation is because it goes against the original motivation of considering sparse binary feedback, in addition to making the experiment design more complicated and comparison to baselines more challenging. We added a comment to Section 6 in order to clarify this.
> - **Lack of Detailed Feedback**
>
>   Even though we did not evaluate it experimentally, it is possible to provide a textual description in order to steer the feature extraction process. Namely, we use the null prompt for extracting attention features from the feedback images in our experiments. However, one may use an arbitrary prompt for each feedback image, describing for instance the specific aspects they like/dislike. This would significantly complicate the experiment design, as one would have to find a way to automatically generate textual descriptions of the feedback. In addition, it again goes against the aim of reducing the amount of prompt engineering required to achieve desired results by just shifting the prompting to a different stage of the process. We also note that attention masking is another simple extension that could be used to constrain the feedback to certain parts of the image (similar to inpainting with Stable Diffusion). We have added a clarifying comment about this to Section 6.
>
> ---
>
> **References**
>
> [1] Deckers et al. 2023. [Manipulating Embeddings of Stable Diffusion Prompts](https://downloads.webis.de/publications/papers/deckers_2023b.pdf)

---

### Official Review · Reviewer_vmZN · 2023-10-28

**Soundness:** 3 good
**Presentation:** 2 fair
**Contribution:** 3 good
**Rating:** 6
**Confidence:** 4

**Summary:**

This paper proposes a training-free method for text-to-image generation with iterative feedback, which is a novel and useful tool. The FABRIC framework is proposed and experiments are well-designed, showing the effectiveness of the method.

**Strengths:**

1. The paper proposes a very interesting and practically meaningful topic.
2. The method design is reasonable, which utilizes the power of self-attention in Stable Diffusion.
3. Despite this is the first training-free iterative-feedback generation work, it designs interesting and sound experiments.
4. The proposed method has great potential to optimize a lot of tasks based on Stable Diffusion.

**Weaknesses:**

The weakness of the paper mainly lies in writing. It is better to incorporate more method descriptions, including model design and formulations in the main script instead of the appendix.

**Questions:**

I'd like to accept this paper if the writing problem is addressed.

---

> ### Author Response · Authors · 2023-11-17
>
> We thank the reviewer for the excellent summary of our contributions and for the concise, constructive feedback. We agree that certain sections of the writing could be improved and have uploaded a revised version of the paper. In particular, the methods section has been majorly overhauled, hopefully addressing your concerns. We refer to the general comment for a more detailed description of the changes that have been made.

---

> > ### Comment · Reviewer_vmZN · 2023-11-22
> > **Response to the revision**
> >
> > The authors provide a better method description in the revised version and addressed my concerns. I keep my original rating towards this paper.

---

### Official Review · Reviewer_dJNm · 2023-11-02

**Soundness:** 3 good
**Presentation:** 3 good
**Contribution:** 2 fair
**Rating:** 5
**Confidence:** 3

**Summary:**

This paper introduces a novel method to control diffusion models to generate user-preferred images through iterative feedback. This method is based on augmenting the attention module. The proposed method is training-free and model-agnostic (as long as attention plays a core role in the image generation model), and can generate images based on any user preferences by having them provide positive and negative labels of their preference on images.

**Strengths:**

- The proposed technique is model-free and training-agnostic, and is easily applicable to most attention-based image generation methods.

- The proposed technique surpasses baselines and enable existing models to follow preferences reasonably

- Extensive exploration of important parts of the proposed technique: the trade-off between diversity and quality, and the effects of adjusting feedback strength on PickScore.

**Weaknesses:**

- **Limited technical novelty**: While the proposed method is effective in incorporating user feedback, the extension to enabling 'iterative feedback' is rather naive, and the feedback is constrained to binary labels (which the author(s) have acknowledged as a limitation). It would be more interesting to explore more advanced way of users' feedback across multiple rounds, and incorporating other modalities, such as text explanations beyond binary preferences.

- **Lack of human rating in a paper focused on iterative human feedback**: While the author(s) have used reasonable proxy to evaluate the effectiveness of the model in following human preferences, it would strengthen the paper if the author(s) can include some form of user study, given this papers' focus is in incorporating human feedback in the image generation process.

- **Missing discussion to some prior work**: I believe the proposed method has some technical similarity to prompt-based image editing methods, such as instruct-pix2pix [1] and prompt2prompt. [2] While the proposed method is different in the types of feedback and preference investigated, it would be great if the author(s) can systematically compare and survey related techniques that use attention map for feedback and/or image editing. I also have some doubts about whether it is reasonable to claim that the method "outperformed" supervised-learning baselines (HPS), see question below.

*References:*

[1] InstructPix2Pix: Learning to Follow Image Editing Instructions. Tim Brooks*, Aleksander Holynski*, Alexei A. Efros. CVPR 2023

[2] Prompt-to-Prompt Image Editing with Cross Attention Control. Amir Hertz, Ron Mokady, Jay Tenenbaum, Kfir Aberman, Yael Pritch, Daniel Cohen-Or. ICLR 2023.

**Questions:**

- While the paper claims to outperform a supervised-learning baseline (HPS LoRA), it is unclear to me how does HPS relate to PickScore, as they both appear to measure human preference. Would the author(s) please clarify how might they relate to each other? As the models are evaluated on PickScore but LoRA-tuned on HPS.

- How does the method relate to/differ from prompt2prompt and instruct-pix2pix? As stated above, it would be helpful to systematically compare them (and other related prior work) in a table.

---

> ### Author Response · Authors · 2023-11-17
>
> We thank the reviewer for the valuable feedback and suggestions to improve our work. In the following, we would like to address your concerns to the best of our ability and answer any open questions.
> - **Limited technical novelty**
>
>   While we agree that the core conditioning mechanism isn’t novel and has been proposed by the authors of ControlNet [1], the combination with weighted attention and CFG makes for a very versatile and flexible algorithm even beyond the scenarios that were evaluated experimentally. For example, real-valued ratings can be incorporated by varying the feedback strength according to the magnitude of the rating and using it as positive/negative feedback depending on the sign. Textual descriptions as well as masking of certain parts of the image can also be supported by varying the prompt used for extracting attention features or by masking out certain keys and values.
>
>   However, both of these extensions go against the core motivation, which is to alleviate the need for prompt engineering by making the best possible use of sparse binary feedback. In addition, it would have further complicated experiment design, making the results harder to reproduce and more difficult for future work to compare against. This was somewhat unclear from the writing and we have added clarifying comments to Section 6.
> - **Lack of human rating in a paper focused on iterative human feedback**
>
>   We agree with the reviewer that studying human interaction with the system would certainly be insightful. Doing a user study was considered, but we ultimately decided against it due to the challenges involved with the design and execution of such a study. To illustrate: A naive study design would simply let the user try out the system and subsequently have them rate their satisfaction and the perceived quality of generated images. This, however, leaves many variables uncontrolled. To name a few: How much of the perceived quality comes from the base model as opposed to the improvements made by FABRIC? How much does generation time (which is noticeably higher for FABRIC) impact user satisfaction and perceived quality? How much does the user interface influence the results? How big is the learning effect from using the system and how much skill is involved in achieving desirable results?
>
>   As a consequence, if not executed properly, the results of such a study would amount to little more than anecdotal evidence (which we now have added to Section 4, Experiments) and might prove difficult to reproduce due to the number of variables that can impact the final rating. Considering all of this, rather than drawing conclusions from potentially weak evidence, we believe that it is appropriate to leave the empirical analysis of human interaction with FABRIC for future work.
> - **Missing discussion of prior work**
>
>   These two papers, prompt2prompt and instruct-pix2pix, are indeed related to our work, as they use similar techniques but have different goals. We thank the reviewer for pointing this out and have added a paragraph on image editing to Section 5 (Related Work).
> - **Comparing HPS and PickScore**
>
>   The reviewer is correct that HPS and PickScore are very similar and solve the same task, which is human preference estimation. In fact, in the early phases of the project, we were using HPS as the main evaluation metric but decided to replace it with PickScore when that was published since it demonstrated superior accuracy thanks to a larger training set (see the table below), which makes for a more accurate human proxy. We continue using the HPS LoRA as a baseline model explicitly trained to maximize human preference, which is ultimately also what PickScore measures. We note that FABRIC outperforms HPS LoRA whether measured by HPS or PickScore (early experiments used HPS).
>   | Method | #prompts | #choices | LoRA of SD available |
>   |---|---|---|---|
>   | HPS | 25k | 25k | yes |
>   | PickScore | 38k | 584k | no |
> - **Systematic comparison of related work**
>
>   We like the idea of adding a systematic comparison of methods which incorporate human feedback in the generation process of diffusion models, even beyond just the techniques using attention-injection, but unfortunately we couldn’t find space to include it in the paper. Instead, we’ll just attach it here and possibly add it to the Appendix if it is considered a valuable addition. The table is provided in the next comment due to the character limit.
>
> Finally, we would like to ask for clarification why the reviewer, despite rating the soundness and presentation of the work as good and the contribution as excellent, ultimately deems the paper to be below the acceptance threshold. Are there significant flaws that prevent acceptance of the paper in its current state but that could be addressed or is there a more fundamental issue?
>
> ---
>
> **References**
>
> [1] https://github.com/Mikubill/sd-webui-controlnet/discussions/1236

---

> > ### Author Response · Authors · 2023-11-17
> >
> > | Method | Iterative | Text-based feedback | Training-free | Sparse/binary feedback | Generate diverse images |
> > |---|---|---|---|---|---|
> > | Textual inversion embeddings | no | no | no | yes | yes |
> > | prompt2prompt | no | yes | yes | no | no |
> > | Diffedit | no | yes | yes | no | no |
> > | Imagic | no | yes | no | no | no |
> > | instruct-pix2pix | no | yes | no | no | no |
> > | DPOK | no | no | no | no | yes |
> > | HPS LoRA | no | no | no | no | yes |
> > | ControlNet (reference mode) | no | no | yes | yes | yes |
> > | ZO-RankSGD | yes | no | no | yes | no |
> > | FABRIC | yes | yes* | yes | yes | yes |
> >
> > ---
> >
> > *FABRIC can incorporate textual descriptions for attention feature extraction, but this was not evaluated in the paper.

---

> > ### Comment · Reviewer_dJNm · 2023-11-22
> >
> > Thanks for responding to my review. I apologize for making an error in the Contribution rating, which is my fundamental issue with this paper (that relates to the overall limitation of technical novelty). I have modified my contribution score accordingly and my overall rating score stays the same.

---

> > > ### Author Response · Authors · 2023-11-23
> > > **Authors' Response to Official Comment by Reviewer dJNm**
> > >
> > > Dear Reviewer,
> > >
> > > Thank you for clarifying your initial rating and clearly stating your fundamental concerns.
> > >
> > > In your initial review, you highlighted specific aspects that contributed towards your impression of limited technical novelty. In particular, you highlighted the following three points:
> > > 1. "the extension to enabling 'iterative feedback' is rather naive"
> > > 2. "the feedback is constrained to binary labels"
> > > 3. "it would be more interesting to explore [...] incorporating other modalities, such as text explanations"
> > >
> > > Regarding the first point, the extension to iterative feedback is a simple but effective approach that fits perfectly with our overall method. Despite its simplicity, in our view, it is the best and most user-friendly technique to iteratively steer the process of image generation in the direction desired by the user, which is the primary goal of FABRIC.
> > >
> > > Additionally, in our previous comments we answered in detail about how your other two suggestions of improvement are already possible in FABRIC.
> > >
> > > > the feedback is constrained to binary labels
> > >
> > > Real-valued ratings are already possible in FABRIC. In particular, they can be incorporated by modifying the weight of each feedback image, which describes how much that specific image should influence future generations. Please refer to our previous comments and to the revised paper, where we have addressed this point in detail.
> > >
> > > > it would be more interesting to explore [...] incorporating other modalities, such as text explanations
> > >
> > > Attaching textual descriptions to each feedback image is already possible in FABRIC. In particular, it can be done by using a user-provided prompt during the U-Net forward pass of the feedback image. Please refer to our previous comments and to the revised paper, where we have addressed this point in detail.
> > >
> > > Given that the first point of critique is unaddressable as a matter of personal opinion, and given that we have addressed, in detail, the other two points of criticism and have revised the paper to reflect our clarifications, we are left a bit confused by the decision to decrease the Contribution score from 4 (excellent) to 2 (fair), which does not seem to take our answer into account.
> > >
> > > Finally, for the sake of completeness, we would like to re-emphasize our contribution:
> > > 1. We tackle the task of incorporating iterative sparse binary feedback into the generative process of diffusion models, which has been largely unexplored as of yet.
> > > 2. We propose a novel method that achieves this through reference-image conditioning, which is training-free and easily extensible and shows.
> > > 3. We propose various experimental setups to evaluate current and future iterative feedback methods competitively and show that the proposed method outperforms existing baselines on each task.
> > >
> > > In our opinion, binary feedback is a particularly relevant setting, as it is an interesting problem to try to improve as much as possible with minimal feedback information. In practice, feedback data is often binary, underlining the relevance of this setting.

---

### Author Response · Authors · 2023-11-17
**General Comment**

Dear Reviewers

Along with detailed replies to each of your comments, we have uploaded a revised version of the paper that implements all the suggestions and addresses the concerns that were raised as best as possible. We would like to highlight some common points raised by reviewers dJNm and bHFH:
- **Limitation of Binary Feedback**

  Reviewers dJNm and bHFH have noted that the proposed method is constrained to binary feedback labels, and while this is true for the experiments we conducted since we intentionally focus on the setting of sparse feedback, we would like to highlight that this is not a fundamental limitation and that it could be easily implemented on the application side (e.g. by scaling the feedback strength according to the user rating). We have added a comment elaborating on this to Section 6 (Future Work and Limitations).
- **Textual Descriptions**

  In the same vein, there is an easy way to provide textual descriptions of the feedback along with the label, which could be used to guide the attention feature extraction process and enable more fine-grained control over the effect of the feedback images. This is also elaborated on in Section 6.

We have also improved the section on related work by adding prompt2prompt and instruct-pix2pix, as suggested by Reviewer dJNm, and have moved it towards the end of the paper due to its growing length (now Section 5). Further, clarifying comments have been added on the relation between HPS and PickScore (Section 4.1) as well as on improving the diversity of generated images (Section 6).
Finally, as suggested by Reviewer vmZN, Section 2 (FABRIC) has been revised considerably with more details and better clarity, as well as an improved figure to illustrate the entire method, including the conditioning mechanism.

We thank all of the reviewers for their insightful comments and suggestions, and we believe that this collaboration has served greatly to improve the paper. We are also open to further suggestions if there are ways for us to continue improving the quality of our work.

---

### Meta-Review · Area_Chair_4TMZ · 2023-12-08

**Metareview:**

This paper presents an approach to control diffusion models to generate a desired image by allowing multiple rounds of binary feedback by giving positive and negative examples from the previous generations. The reviewers found the approach interesting and the iterative refinement workflow enabled by the work has a good potential for "Personalized Content Creation" (Reviewer bHFH). On the technical side, Reviewer dJNm noted that "The proposed technique is model-free and training-agnostic, and is easily applicable to most attention-based image generation methods."

That said, all the reviewers have reservations with the paper. One major issue that stands out is limited feedback information. The work only allows binary feedback, yet more nuanced is often needed for refining image generation. Reviewer dJNm pointed out "While the proposed method is effective in incorporating user feedback, the extension to enabling 'iterative feedback' is rather naive." Reviewer bHFH shared the same concern: "The current feedback collection method only allows users to provide binary preferences (like/dislike) for the images. This limitation prevents users from providing nuanced feedback about specific aspects of an image."

Reviewer dJNm also pointed out a crucial limitation of the work, which is the lack of human rating. The authors responded by explaining such a user eval is difficult to perform. It might be the case that quantitative measures can be challenging. However, even observations or qualitative findings can be valuable, which will greatly enhance the paper. Reviewer bHFH pointed a number of issues that the current approach might encounter, such as "Limited Expansion of Distribution" and "Diversity Collapse".

**Justification For Why Not Higher Score:**

The paper suffers a number of issues as detailed above.

**Justification For Why Not Lower Score:**

The overall approach is interesting and has a potential to enable personalized image generation.

---

### Decision · Program_Chairs · 2024-01-16

Reject